# Impact of a Multiplex Polymerase Chain Reaction Assay on the Clinical Management of Adults Undergoing a Lumbar Puncture for Suspected Community-Onset Central Nervous System Infections

**DOI:** 10.3390/antibiotics9060282

**Published:** 2020-05-26

**Authors:** Matthew A. Moffa, Derek N. Bremmer, Dustin Carr, Carley Buchanan, Nathan R. Shively, Rawiya Elrufay, Thomas L. Walsh

**Affiliations:** 1Division of Infectious Diseases, Allegheny General Hospital, Allegheny Health Network, 320 East North Avenue 4th Floor East Wing, Suite 406, Pittsburgh, PA 15212, USA; nathan.shively@ahn.org (N.R.S.); rawiya.elrufay@ahn.org (R.E.); thomas.walsh@ahn.org (T.L.W.); 2Division of Infectious Diseases, West Penn Hospital, Allegheny Health Network, 4800 Friendship Ave, Pittsburgh, PA 15224, USA; 3Department of Pharmacy, Allegheny General Hospital, Allegheny Health Network, 320 East North Avenue 4th Floor East Wing, Suite 406, Pittsburgh, PA 15212, USA; derek.bremmer@ahn.org (D.N.B.); dustin.carr@ahn.org (D.C.); 4Department of Pharmacy, West Penn Hospital, Allegheny Health Network, 4800 Friendship Ave, Pittsburgh, PA 15224, USA; carley.buchanan@ahn.org

**Keywords:** lumbar puncture, meningitis, encephalitis, herpes simplex virus, polymerase chain reaction

## Abstract

Patients admitted from the community with a suspected central nervous system (CNS) infection require prompt diagnostic evaluation and correct antimicrobial treatment. A retrospective, multicenter, pre/post intervention study was performed to evaluate the impact that the BioFire^®^ FilmArray^®^ meningitis/encephalitis (ME) panel run in-house had on the clinical management of adult patients admitted from the community with a lumbar puncture (LP) performed for a suspected CNS infection. The primary outcome was the effect that this intervention had on herpes simplex virus (HSV) polymerase chain reaction (PCR) turnaround time (TAT). Secondary outcomes included the effect that this intervention had on antiviral days of therapy (DOT), total antimicrobial DOT, and hospital length of stay (LOS). A total of 81 and 79 patients were included in the pre-intervention and post-intervention cohorts, respectively. The median HSV PCR TAT was significantly longer in the pre-intervention group (85 vs. 4.1 h, *p* < 0.001). Total antiviral DOT was significantly greater in the pre-intervention group (3 vs. 1, *p* < 0.001), as was total antimicrobial DOT (7 vs. 5, *p* < 0.001). Pre-intervention hospital LOS was also significantly longer (6.6 vs. 4.4 days, *p* = 0.02). Implementing the ME panel in-house for adults undergoing an LP for a suspected community-onset CNS infection significantly reduced the HSV PCR TAT, antiviral DOT, total antimicrobial DOT, and hospital LOS.

## 1. Introduction

Central nervous system (CNS) infections, such as meningitis and encephalitis, are infectious disease emergencies. Patients suspected of having a CNS infection require urgent diagnostic evaluation with a lumbar puncture (LP) and cerebrospinal fluid (CSF) analysis, along with prompt antimicrobial therapy. In the United States, mortality from bacterial meningitis ranges from 15%–25% [1]. Thus, the ability to rapidly diagnose a pathogen is critical to ensure effective therapy.

Often, patients that undergo CSF analyses remain on broad antimicrobial coverage while waiting days for diagnostic results [2]. This delay can result in preventable harm from unnecessary drug exposure, as well as prolonged hospital stays. Indeed, most LPs performed for suspected CNS infections subsequently reveal no evidence of infection [3]. Thus, the vast majority of patients that undergo an LP for a suspected CNS infection are exposed to excess antimicrobials that can lead to unintended consequences such as antimicrobial drug resistance and *Clostridioides difficile* infection (CDI) [4,5]. Similarly, each additional day spent in a hospital increases the risk of developing a hospital-acquired infection and represents extra cost to the healthcare system [6].

Our microbiology laboratory previously sent CSF samples to a reference lab for viral polymerase chain reaction (PCR) testing, including herpes simplex virus (HSV), varicella zoster virus (VZV), enterovirus, and cytomegalovirus (CMV) PCR assays. This process would result in significant reporting delays while patients remained in the hospital on empiric antimicrobial coverage. In order to expedite the turnaround time (TAT) of CSF studies, our laboratory implemented a multiplex PCR assay in-house with the BioFire^®^ FilmArray^®^ Meningitis/Encephalitis (ME) panel. This assay is able to detect six bacteria: *Escherichia coli* K1, *Haemophilus influenzae*, *Listeria monocytogenes*, *Neisseria meningitidis*, *Streptococcus agalactiae*, *Streptococcus pneumoniae*; seven viruses: CMV, enterovirus, HSV-1, HSV-2, human herpesvirus 6, human parechovirus, VZV; and one yeast: *Cryptococcus neoformans/gattii.* The objective of this study was to evaluate the impact this rapid diagnostic assay had on patient care, including HSV PCR TAT, antimicrobial days of therapy (DOT), and hospital length of stay (LOS).

## 2. Results

A total of 503 patient charts were screened in the pre-intervention group, while 218 were screened in the post-intervention group (Figure 1). After application of the study criteria, a total of 81 and 79 patients were included in the pre-intervention and post-intervention groups, respectively. Baseline demographics can be seen in Table 1. There were significantly more females in the post-intervention group. There were no significant differences in age, race, immunocompromised state, ICU admissions, 30-day mortality, CSF findings, or rates of antimicrobial pretreatment prior to LP.

Detected pathogens can be found in Figure 2. Five patients (6.1%) in the pre-intervention group had a positive PCR assay for either HSV or VZV, compared to six patients (7.6%) in the post-intervention group. Neither group had a positive CSF culture, CSF bacterial antigen assay, CSF bacterial PCR assay, or a positive blood culture with non-commensal bacteria.

The median HSV PCR TAT was significantly longer in the pre-intervention group, 85 vs. 4.1 h, *p* < 0.001 (Table 2). Similar findings were seen with other viral PCR TATs, including VZV, enterovirus and CMV. Correspondingly, the median total antimicrobial DOT was significantly greater in the pre-intervention group (7 vs. 5, *p* < 0.001). The median total antiviral DOT was also significantly greater in the pre-intervention group (3 vs. 1, *p* < 0.001). In the subset of patients without a positive HSV or VZV PCR, pre-intervention antiviral DOT remained significantly greater as well (2 vs. 1, *p* < 0.001). Lastly, hospital LOS was significantly longer in the pre-intervention group (6.6 vs. 4.4 days, *p* = 0.02).

## 3. Discussion

The goal of implementing the ME panel in-house for adult patients admitted from the community with a suspected CNS infection was to provide clinicians with rapid, actionable results. Knowing the pathogen in a CSF infection within hours gives one the ability to expeditiously optimize antimicrobial therapy. In turn, we postulated that reducing the TAT of viral PCR assays would have positive downstream effects, such as less empiric antimicrobial exposure as well as less time spent in the hospital.

By implementing the ME panel in-house, this study found a significant reduction in HSV PCR TAT. The reduction in diagnostic TAT by more than three days led to a significantly shortened duration of antimicrobial therapy and hospital LOS. This study adds to the growing body of literature that links the implementation of this assay with a reduction in hospital LOS [7]. While we did not analyze cost savings from this intervention, prior published data have suggested a cost-benefit by utilizing multiplex PCR testing on all adult patients from the community with a suspected CNS infection. [8] Our findings of reduced antimicrobial DOT and reduced hospital LOS would support this. We did find that the ME panel was sent 28 times after an LP was performed for a non-infectious indication, such as patients undergoing intrathecal chemotherapy or CSF shunt placement. This led to targeted education by our antimicrobial stewardship team to avoid unnecessary costs.

While a patient admitted from the community with a suspected CNS infection represents a medical emergency, it is common not to find a pathogen causing a treatable infection in the majority of patients, similar to our findings [3,9,10]. A rapid, multiplex PCR not only provides clinicians with the ability to quickly adjust therapy to target a pathogen, but also provides the ability to rapidly discontinue unnecessary antimicrobials. Adult patients admitted from the community with a suspicion of meningitis and encephalitis are routinely placed on three to four antimicrobials, per national guidelines [11,12]. Diagnostic delays can subject patients to the unintentional harm of excessive antimicrobial exposure, such as antimicrobial resistance and CDI [4,5]. Antimicrobial exposure can also lead to adverse drug events such as acute kidney injury. Although we did not evaluate the incidence of renal injury in this study, acute kidney injury from acyclovir may occur within two days of therapy [13]. We found a reduction in antiviral DOT from three to one after implementation of the ME panel, which may help limit the potential harm from acyclovir use. Shorter acyclovir therapy may also be advantageous during national drug shortages so distribution can be allocated to those in greatest need. Lastly, the demonstrated reduction in hospital LOS may help mitigate risk of developing a hospital-acquired infection and reduce healthcare costs [6].

This is one of the first studies to evaluate the clinical impact of an in-house multiplex PCR assay on the care of adult patients admitted from the community with a suspected CNS infection. Similar to prior studies, we found a significantly reduced diagnostic TAT and a reduction in antiviral therapy [2,7,9,10,14]. Additionally, our study included a diverse patient population by incorporating both a quaternary academic referral center as well as a community-based teaching hospital.

There are several important limitations in our study, most importantly being the retrospective design. While we did not calculate a disease severity score for the two groups, we captured rates of ICU admissions and 30-day mortality rates to provide a broad overview of illness severity, which was similar among both groups. Interestingly, there were no bacterial pathogens isolated in either group. It is unclear if this is due to the high rates of antimicrobial pretreatment prior to LP seen equally in both groups. This assay has a reported overall sensitivity of 94.2% and specificity of 99.8% [15]. Thus, one would expect a low number of false-positive and false-negative results. Further research should evaluate whether our findings would be replicated in a setting with higher rates of bacterial meningitis. Even in such settings, our findings in cases without a treatable pathogen remain impactful for the role of antimicrobial stewardship.

In conclusion, we found that implementing an in-house multiplex PCR assay for adults undergoing an LP for a suspected community-onset CNS infection significantly reduced the HSV PCR TAT, antimicrobial DOT, and hospital LOS. Such an assay may be useful in assisting centers looking to reduce diagnostic TAT, promote antimicrobial stewardship, and control healthcare costs.

## 4. Materials and Methods

### 4.1. Study Design

We conducted a retrospective, multicenter, pre/post intervention study evaluating the impact that the ME panel had on the clinical management of adult patients admitted from the community with an LP performed for a suspected CNS infection. The pre-intervention time period was from October 2016 to September 2017, while the post-intervention time period was from October 2017 to September 2018.

### 4.2. Study Setting

Allegheny General Hospital (AGH) is a 631 bed quaternary care teaching facility with approximately 22,000 inpatient admissions yearly. West Penn Hospital (WPH) is a 317 bed community based teaching hospital with nearly 6800 inpatient admissions annually. Both facilities are located in Pittsburgh, Pennsylvania, and are members of the Allegheny Health Network (AHN). The study was approved by the AHN Institutional Review Board as a quality assurance/quality improvement project, study code 2018-112.

### 4.3. Intervention

In October 2017, our core microbiology laboratory at AHN implemented a multiplex PCR assay in-house with the BioFire^®^ FilmArray^®^ ME panel run immediately upon receipt 24 h per day, 7 days per week. This assay was performed as per the manufacturer’s instructions. Prior to launch, a memo was distributed to all medical staff providing background information regarding this instrument, including availability, appropriate ordering, targeted pathogens, and expected TAT. The decision on which diagnostic tests, including the ME panel, to order in both the pre- and post-intervention groups was left to the discretion of the treating physician. Prior to the intervention, individual viral PCR assays, including HSV, were sent out to reference labs. No other interventions pertaining to CNS infections were performed during either study period.

### 4.4. Data Collection

In the pre-intervention group, potential candidates were identified by screening all patients that had a CSF culture performed during the designated time period. In the post-intervention group, potential candidates were identified by screening all patients that had the ME panel study performed during the designated time period. We did not screen all patients with a CSF culture in the post-intervention group as we were only interested in identifying those who had a ME panel performed to study the impact of the assay. Data were extracted from the electronic medical record (EMR) by study investigators, which included baseline demographics, laboratory and CSF studies, and antimicrobial prescriptions both inpatient and if prescribed for outpatient use.

Patients were included if they were ≥ 18 years of age and admitted from the community with an LP performed for a suspected CNS infection. Patients were excluded for the following reasons: LP performed for indication other than suspected CNS infection; LP performed after hospital day 3; patient at an outside hospital for more than 48 h prior to transfer to our facility; patient on a systemic antimicrobial for a non-CNS indication; unable to determine duration of therapy; neurosurgical patient and/or CNS device infection; fungal CNS infection; ME panel not sent during time of LP in post-intervention group.

### 4.5. Study Outcomes and Definitions

The primary outcome of this study was to determine the effect that an intervention of an in-house ME panel multiplex PCR had on HSV PCR TAT. Secondary outcomes included the effect this intervention had on VZV PCR TAT, enterovirus PCR TAT, CMV PCR TAT, total antimicrobial DOT, total antiviral DOT, total antiviral DOT in patients with a negative HSV or VZV PCR, and hospital LOS.

TAT was defined as the time between laboratory collection and laboratory result display in the EMR. DOT was defined as the aggregate sum of days for which any amount of a specific antimicrobial agent was administered as documented in the EMR and/or prescribed for outpatient use. Antiviral therapy included intravenous acyclovir and oral valacyclovir. Antimicrobial therapy included antiviral therapy and any antibacterial agent prescribed for a suspected CNS infection. Patients were considered immunocompromised if they had any of the following: absolute neutrophil count (ANC) ≤ 500 cells/µL; human immunodeficiency virus (HIV) with CD4 ≤ 200 cells/µL; receipt of chemotherapy within previous 2 weeks for active malignancy; administration of immunosuppressive agents, including administration of corticosteroid dose equivalent to 20 mg prednisone for at least 1 month; prior solid organ or stem cell transplant recipient. A traumatic LP was defined as CSF red blood cells ≥ 500 cells/µL.

### 4.6. Statistical Analysis

Based on prior published data of reduced HSV PCR TAT using a rapid PCR test in-house, a priori power analysis was performed for a Wilcoxon rank-sum test [16]. A sample size of 76 patients in each arm was necessary to achieve 80% power with an alpha of 0.05. Categorical outcome data were compared using the chi-square test or Fisher exact test and continuous outcome data were compared using the Wilcoxon rank-sum test or *t* test depending on distribution. A *p* value of < 0.05 was considered statistically significant for all outcome data. Statistical analysis was performed using R, version 3.5.1 (R Foundation for Statistical Computing, Vienna, Austria).

## 5. Conclusions

Implementing a multiplex PCR in-house with the BioFire^®^ FilmArray^®^ ME panel for adults undergoing an LP for a suspected community-onset CNS infection significantly reduced the HSV PCR TAT, antimicrobial DOT, and hospital LOS. This intervention can assist healthcare centers with diagnostic TAT reduction, promotion of antimicrobial stewardship, and controlling healthcare costs.

## Figures and Tables

**Figure 1 antibiotics-09-00282-f001:**
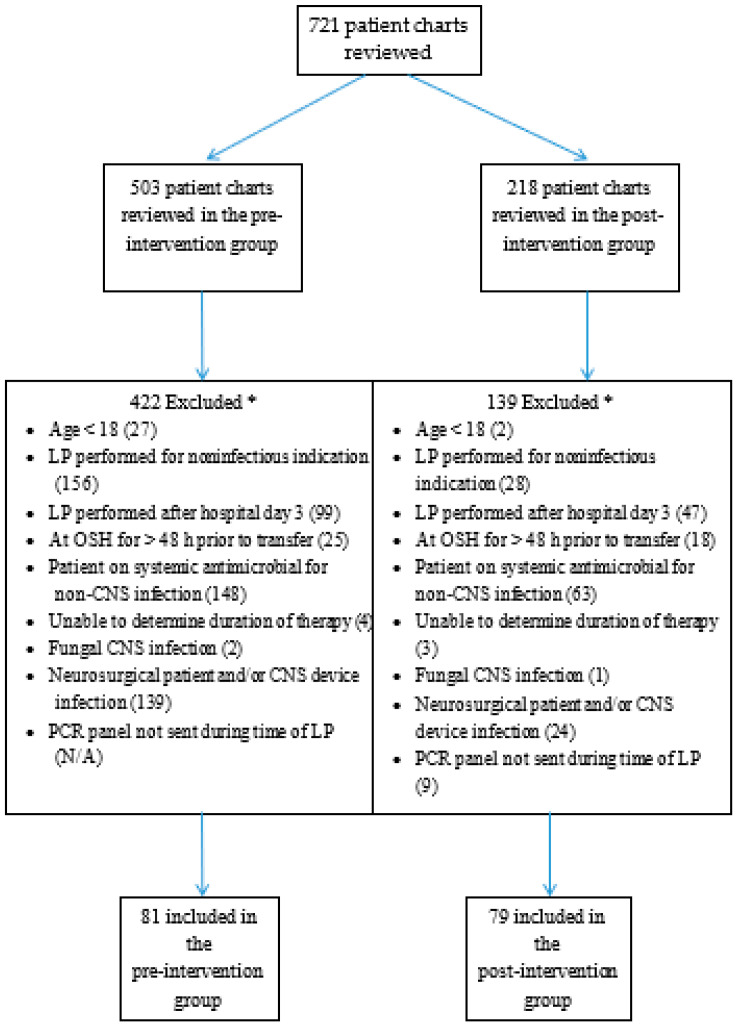
Patient Screening. * Multiple exclusion criteria could be applied to each patient. Abbreviations: LP: Lumbar puncture, OSH: Outside hospital, CNS: Central nervous system, PCR: Polymerase chain reaction.

**Figure 2 antibiotics-09-00282-f002:**
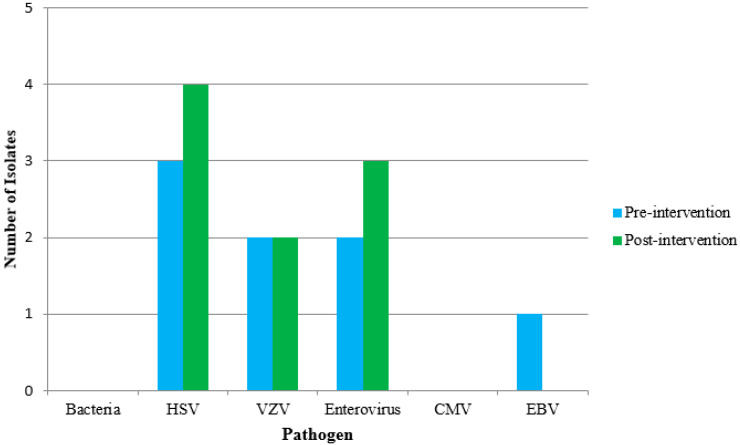
Detected Pathogens. Abbreviations: HSV: Herpes simplex virus, VZV: Varicella-zoster virus, CMV: Cytomegalovirus, EBV: Epstein-Barr virus.

**Table 1 antibiotics-09-00282-t001:** Demographics and Baseline Data.

	Pre-Intervention (*n* = 81)	Post-Intervention (*n* = 79)	*p* Value
Age, years, mean ± SD	50.6 ± 20.1	49.9 ± 17.5	0.8
Female sex, *n* (%)	35 (43.2)	49 (62)	0.02
Race	−	−	0.10
White, *n* (%)	60 (74.1)	63 (79.7)	−
African American, *n* (%)	18 (22.2)	9 (11.4)	−
Other/unknown, *n* (%)	3 (3.7)	7 (8.9)	−
ICU admission, *n* (%)	26 (32.1)	19 (24.1)	0.26
30-day mortality, *n* (%)	5 (6.2)	3 (3.8)	0.72
Peripheral WBC, k/µL, mean ± SD	9.6 ± 4.2	10.5 ± 6.1	0.29
CSF WBC, per µL, mean ± SD	61 ± 157	162 ± 798	0.27
CSF neutrophils, %, mean ± SD	21 ± 31	18 ± 29	0.51
CSF lymphocytes, %, mean ± SD	58 ± 33	58 ± 34	0.94
CSF glucose, mg/dL, mean ± SD	80 ± 38	76 ± 26	0.37
CSF protein, mg/dL, mean ± SD	57 ± 34	60 ± 46	0.56
Traumatic LP, *n* (%)	8 (9.9)	6 (7.6)	0.61
Antimicrobial pretreatment prior to LP, *n* (%)	36 (44.4)	33 (41.8)	0.73

Abbreviations: SD: Standard deviation, ICU: Intensive care unit, WBC: White blood cell count, CSF: Cerebrospinal fluid, LP: Lumbar puncture.

**Table 2 antibiotics-09-00282-t002:** Primary and Secondary Outcomes.

	Pre-Intervention (*n* = 81)	Post-Intervention (*n* = 79)	*p* Value
HSV PCR TAT, hours, median (IQR)	85 (77–99.6)	4.1 (2.9–5.4)	<0.001
VZV PCR TAT, hours, median (IQR)	124.1 (102–152.3)	4.1 (2.9–5.4)	<0.001
Enterovirus PCR TAT, hours, median (IQR)	112.4 (87.2–140.2)	4.1 (2.9–5.4)	<0.001
CMV PCR TAT, hours, median (IQR)	98.4 (79.6–123.4)	4.1 (2.9–5.4)	<0.001
Total antiviral DOT, median (IQR)	3 (1–5)	1 (0–2)	<0.001
Total antiviral DOT if negative HSV/VZV PCR, median (IQR)	2 (1–5)	1 (0–2)	<0.001
Total antimicrobial DOT, median (IQR)	7 (4–13)	5 (1–9.5)	<0.001
LOS, days, mean ± SD	6.6 ± 7.6	4.4 ± 3.5	0.02

Abbreviations: HSV: Herpes simplex virus, PCR: Polymerase chain reaction, TAT: Turnaround time, IQR: Interquartile range, CMV: Cytomegalovirus, VZV: Varicella-zoster virus, LOS: Length of stay, DOT: Days of therapy, SD: Standard deviation.

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
