# Peer review of "Impact of a Multiplex Polymerase Chain Reaction Assay on the Clinical Management of Adults Undergoing a Lumbar Puncture for Suspected Community-Onset Central Nervous System Infections"

_antibiotics, 2020, doi:10.3390/antibiotics9060282_

Round 1
Reviewer 1 Report
The study evaluates the impact of an in house multiplex PCR on clinical management of adults undergoing LP for suspected CNS infection. CSF analysis takes days for complete analysis and during this phase patients are usually treated with broad antibiotics which increases the chance of antibiotic resistance and some times the prolonged stay may lead to secondary infection making the situation critical. With the use of the in house multiplex PCR there is a significant reduction in HSV PCR turnaround time , antimicrobial and antiviral days of therapy making the whole process faster and probably less stressful.
Reviewer 2 Report
Overall, this article on the impact of a multiplex PCR assay on management of adults with suspected community onset CNS infections was well written and easy to read. Please see below for minor suggestions for improvement.
- Consider adding what viral & bacterial organisms are tested for with the ME panel in the introduction so readers that may be less familiar with the test are aware.
- Consider adding sensitivity/specificity data into the discussion and how that may play a role in false positives/negatives and potentially affect management of patients.
- Since the last sentence of your conclusion says, "Such an assay may be useful to assist centers looking to reduce time to optimal therapy, promote antimicrobial stewardship, and control healthcare costs," consider adding time to optimal therapy as an outcome. Otherwise, this cannot be concluded from your study. I do think this would be a good addition if you have the data.
- Please clarify what "LP indicated for non-infectious reasons" entails. Why were ME panels being done if infection wasn't suspected? I also recommend discussing this in the discussion as it shows that there is inappropriate use of the test which could lead to excess costs and perhaps additional education is needed for providers on when to order the test.
- Regarding the intervention, please clarify if there was any other intervention being done during either time period. For example, did the antimicrobial stewardship team set up alerts for positive or negative tests and make active interventions?
- Consider discussing how shorter therapy courses of antivirals may be particularly beneficial in situations like the most recent acyclovir shortage.
- Grammar: line 105 - There is an extra "to" in this sentence.
Reviewer 3 Report
Dear authors,
The submitted article, "Impact of a Multiplex Polymerase Chain Reaction Assay on the Clinical Management of Adults Undergoing a Lumbar Puncture for Suspected Community-Onset Central Nervous System Infections", is a very well written and concise study with valuable findings. However, there are various shortcomings in this paper that need to be addressed, in particular, the introduction and discussion need to be improved. See my comments below:
- Is it common practice in this journal to have titles after author names?
- Line 22 suggestion, “require prompt antimicrobial treatment and diagnostic evaluation”, change to “require prompt diagnostic evaluation and correct antimicrobial treatment”.
- Line 23, the term “in-house” in reference to a diagnostic assay typically implies that this assay was designed by the research group, should you make reference to the BioFire® FilmArray® Meningitis/Encephalitis (ME) panel multiplex PCR assay here?
- Line 26, “The primary outcome was herpes simplex virus (HSV) PCR turnaround time (TAT).” This line could be significantly improved. What exactly are you referring to? As in, the primary outcome was to determine what effect implementing an in-house multiplex PCR assay into the routine diagnostic procedure had on HSV PCR TAT? Should expand this.
- Line 27, “Secondary outcomes included antiviral…” this sentence could also be improved for the same reasons as per line 26.
- Line 28/29, what exactly does “intervention” mean in “pre-intervention” and “post-intervention”. This needs to be made clearer here. After reading the whole paper it is clear that pre-intervention is prior to implementation of the multiplex PCR assay and post-intervention is after implementation of the multiplex PCR assay, however, this is not clear in the early sections of the paper. Also, you could give more details on the diagnostic approach prior to implementation of the ME panel assay.
- References are needed for the information in lines 39 to 42.
- Line 52, “our microbiology laboratory sent out CSF samples to a reference lab..” far too informal, state the details of mirco labs in question here and maybe expand on the testing carried out.
- The connection between the last and second last paragraph in the introduction is very poor. You have introduced to the reader why delay in diagnosis is bad, but then you should introduce PCR and multiplex PCR, and explain why these methods improve diagnosis etc. The introduction is very short for a full article and could do with this extra content especially considering evaluating the effects of implementing a multiplex PCR assay is the focus of this paper.
- Line 59, “503 charts”, need to explain what a “chart” is. Author is assuming knowledge on behalf of the reader.
- Why were over double the number of charts screened in the pre-intervention group compared to the post-intervention group?
- What do you mean by “isolated pathogens”? Are these just detected pathogens or do you mean isolated and cultured?
- In Figure 2, it shows that EBV is identified by the pre-intervention approach but not by the post-intervention approach. Why is this? This is not discussed or mentioned anywhere in the paper.
- The start of the discussion is poor. It is typical for the opening paragraph of a discussion to briefly summaries the project/aims before discussing the results/outcomes.
- Line 101, “prior published data has suggested a cost-benefit by testing all adult patients” you should make it clearer here that by “testing” you mean using the multiplex PCR assay.
- Line 105, “treatable pathogen” reconsider phrasing, you are treating the infection caused by the pathogen.
- Line 116, “our study has important strengths”, would remove this line, not impartial or objective.
- Line 149, does info need to be given on how the ME panel was used? As in, was the multiplex PCR assay carried out as per manufacturer’s instructions etc?
- Was ethical approval granted for this study?
- Line 152, “The decision on which studies, including the ME panel, to order…” can you expand what you mean by “studies” here?
- Line 154, “individual viral PCR assays, including HSV, were sent out to reference labs.” Does more info need to be given on the type of PCR analysis done at these reference labs? Were the same pathogens test as per the ME panel? Was multiplex PCR performed here? Is the only reason that implementing the in-house ME panel multiplex PCR improved outcomes a result of the fact that testing could now be done in-house and not need to be sent out, thus saving time?
- Line 156, “In the pre-intervention group, patients were…”, the word “patients” here should probably be changed to “potential candidates”?
- Line 170, opening line could be improved, i.e. “The primary outcome of this study was to determine the impact of implementing the ME panel multiplex PCR assay on the existing diagnostic procedure, and in particular, the effect this had on HSV PCR TAT etc”. This is by no means a perfect alternative but this sentence needs to be improved. Sometimes the info in this paper is very sparse.
